# A Long Way to Go: Investigating Length Correlations in RLHF

**Prasann Singhal**
The University of Texas at Austin
prasanns@cs.utexas.edu

**Tanya Goyal**
Princeton University
tanyagoyal@princeton.edu

**Jiacheng Xu**
Salesforce AI
jiacheng.xu@salesforce.com

**Greg Durrett**
The University of Texas at Austin
gdurrett@cs.utexas.edu

## Abstract

Great success has been reported using Reinforcement Learning from Human Feedback (RLHF) to align large language models, with open preference datasets enabling wider experimentation, particularly for "helpfulness" in tasks like dialogue and web question answering. Alongside these improvements, however, RLHF also often drives models to produce longer outputs. This paper demonstrates, on three diverse settings, that optimizing for response length is, much more than previously thought, a significant factor behind RLHF. Studying the strategies RL optimization uses to maximize reward, we find improvements in reward to largely be driven by increasing response length, instead of other features. Indeed, we find that even a *purely* length-based reward reproduces most downstream RLHF improvements over supervised fine-tuned models. Testing a comprehensive set of length-countering interventions, we identify the dominant source of these biases to be reward models, which, by studying training dynamics, we find are non-robust and easily influenced by length biases in preference data.

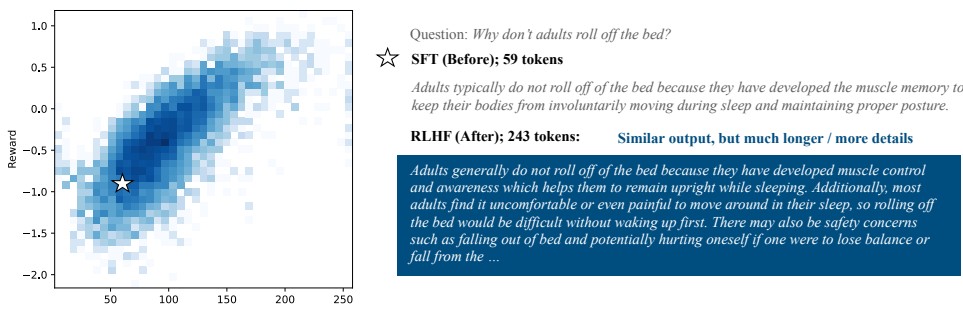

Figure 1: Log-scaled heatmap of lengths of SFT outputs vs. learned reward model scores for WebGPT (left). The graph shows that reward scores are strongly correlated with length. RLHF with these consistently leads to longer outputs (right).

## 1 Introduction

Reinforcement Learning from Human Feedback (RLHF) is widely used to align large language models (LLMs) with desired downstream properties such as helpfulness or harmlessness (Ouyang et al., 2022; Bai et al., 2022). This procedure entails (1) learning a reward model to estimate human preferences and (2) using RL to maximize that reward. Its success relies on two things. First, the reward model must be correctly specified and not misaligned with human preferences (Zhuang & Hadfield-Menell, 2021; Pang et al., 2022; Bobu et al., 2023). Second, the optimization algorithm must successfully maximize reward without straying too far from a sensible initial distribution.

Not meeting these conditions can lead to over-optimization of the reward model at the expense of human judgments (Dubois et al., 2023) or pathological "reward hacking" (Skalse et al., 2022). Ad hoc adjustments (Touvron et al., 2023b; Zheng et al., 2023b) to PPO have stabilized the process, but there is still little work examining what features improve in policy models, and to what extent these correspond to meaningful improvements in quality versus optimizing shallow reward correlations (Pang et al., 2022).

In this work, we focus on one such feature: output length. Prior work in RLHF has noted that the length of sampled outputs increases after RLHF (Stiennon et al., 2020; Nakano et al., 2021; Dubois et al., 2023; Zheng et al., 2023b; Sun et al., 2023; Wu et al., 2023) but largely dismissed it as an artifact of PPO. In this paper, we study this phenomenon in depth to understand its underlying causes, and to what extent length is a meaningful feature vs. a spurious correlation that PPO optimizes for.

We organize our main findings into three parts. First, we compare vanilla PPO and SFT models in three diverse settings (WebGPT (Nakano et al., 2021), Stack (Lambert et al., 2023) and RLCD Yang et al. (2023)) and show that length dominates reward optimization in PPO to a surprising extent. In fact, for two settings, we find that the **PPO improvements disappear if we restrict our comparison to similar length outputs from PPO and SFT**.

We then run a controlled experiment with the PPO pipeline, but replacing learned reward models with a purely length-based heuristic score. We find that PPO performance with this length-only reward is close to standard PPO (56% vs 58% win-rate of standard PPO on WebGPT and 64% vs 63% win-rate of standard PPO on RLCD). This **shows shortcomings of current reward models, which fail to convincingly outperform simple heuristics**, but more broadly questions the recent "progress" reported with popular metrics.

Exploring a comprehensive set of anti-length interventions, including changing preference datasets, reward model training, policy rollout strategies, reward scores and KL loss, we find that no strategy works for all settings. However, we find that interventions generally brings length closer to the base model, sometimes without degrading performance.

Output length may be a legitimate feature to optimize for, as it may correspond to greater informativeness; however our results hint at a concerning scenario wherein PPO struggles to improve reward without increasing length, failing optimize for important non-length features. We find that this is because learned reward models themselves exhibit very strong correlations with length (see Figure 1) at the cost of other features, from training itself.

Taken together, our findings: (1) show that current reward models only model shallow aspects of human preferences; (2) call into question PPO "improvements" using reward models on the datasets we study; and (3) show what interventions are effective and call for better preference data and downstream evaluation.

## 2 Background and Task Setup

Text generation models (Sutskever et al., 2014; Bahdanau et al., 2015) place a distribution over target output $\mathbf{y} = (y_1, \ldots, y_n)$ given input sequences of tokens $\mathbf{x}$, i.e., $\pi_\theta$: $p(\mathbf{y} \mid \mathbf{x}; \pi_\theta) = \prod_{k=1}^{n} p(y_k \mid \mathbf{y}_{<k}, \mathbf{x}; \pi_\theta)$. Typically, these models are trained with both language modeling pre-training (learning to predict the next word given context) and supervised fine-tuning (SFT; learning to generate outputs to maximize the likelihood of references on some dataset).

**RLHF** can then be broken into two stages. First, preference data of the form $P = \{(x_1, y_1^+, y_1^-), \ldots, (x_n, y_n^+, y_n^-)\}$ is collected, where $x_i$ is the prompt, $y_i^+$ is the preferred continuation, and $y_i^-$ is the dispreferred continuation. This dataset is used to train a scalar **reward model** $R(x, y)$ such that for any preference, $R(x_i, y_i^+) > R(x_i, y_i^-)$. We use the standard Bradley-Terry preference model (Bradley & Terry, 1952), that is, $P(y_1 > y_2 \mid x) = \frac{\exp(R(x,y_1))}{\exp(R(x,y_1)) + \exp(R(x,y_2))}$, to maximize the log likelihood of the observed preferences.

Second, **proximal policy optimization (PPO)** (Schulman et al., 2017) is used further train the SFT model ($\pi_\theta^{\text{SFT}}$) to get $\pi_\theta^{\text{RL}} = \text{PPO}(\pi_\theta^{\text{SFT}}, R)$. This training maximizes the reward

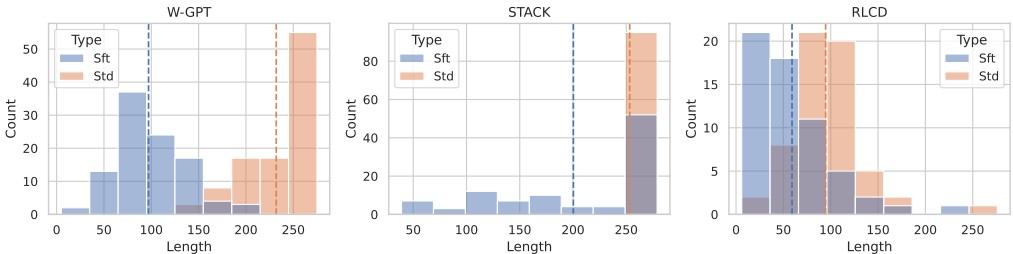

Figure 2: Output lengths of SFT model before (blue) and after (red) standard PPO (STD); averages shown with dashed lines. Across all settings, PPO leads to large length increases.

$R(x_i, y_i \sim \pi_\theta(y|x_i))$ on training data $(x_1, \ldots, x_m)$, while not deviating too strongly from the initial distribution. Specifically, it maximizes the following objective, where $\lambda$ controls the strength of a Kullback-Leibler (KL) penalty between SFT and the current policy:

$$R_{\text{ppo}}(x, y) = R(x, y) - \lambda D_{\text{KL}}(\pi_\theta^{RL}(y|x) \| \pi_\theta^{\text{SFT}}(y|x)) \tag{1}$$

## 2.1 Experimental Setup

**Task settings**   We experiments on three popular "helpfulness" preference datasets, ensuring diversity in task settings and source of preference labels (examples in Appendix D):

1. **WebGPT (Question answering; human labels)** Nakano et al. (2021) contains human annotated preference labels for the open-domain long-form question answering task (Fan et al., 2019). It includes 19.6K examples (mean tokens per $y$ = 169).

2. **Stack (Technical question answering; upvotes)** Lambert et al. (2023) contains technical StackExchange questions, with preference labels between two answers derived automatically from the number of upvotes. We use 100K (mean tokens per $y$ = 236) pairs from the dataset following the Hugging Face implementation (von Werra et al., 2020).

3. **RLCD (Multi-turn conversation; synthetic preferences)** Yang et al. (2023) includes multi-turn dialogue data. Starting from input instructions in the Helpful/Harmless dataset by Anthropic (Bai et al., 2022), preferred and not-preferred outputs are automatically generated using prompt heuristics, e.g. appending "generate unhelpful outputs" to the prompt. We use the helpfulness subset (40K preferences; mean tokens per $y$ = 45).

**Implementation Details**   We use Huggingface TRL (von Werra et al., 2020) with hyperparameters we find to work best based on reward and downstream evaluation ($\lambda$ = 0.04, batch size 64; more in Appendix B). We use Llama-7B models (Touvron et al., 2023a) as our base for all experiments, and use LoRA (rank= 16) (Hu et al., 2021) to enable PPO training with limited GPU resources. We choose publicly available SFT models for each setting: AlpacaFarm SFT for WebGPT and RLCD, and TRL SFT for Stack.

**Evaluation Metrics**   We evaluate with (1) (intrinsic) **reward scores** from task-specific reward models used in the PPO process; and (2) (downstream) **AlpacaFarm simulated preferences** (Dubois et al., 2023), popularly used as a proxy for humans in most recent work (Dubois et al., 2023; Zheng et al., 2023a). This queries 12 OpenAI API-based "annotators" to choose between two outputs, and reports pairwise "win rate" of a model over another (e.g. SFT's win-rate over PPO), testing on 500 held-out datapoints for each task. While useful for additional context, we qualify this metric by noting that it itself may have length biases, which we partially examine in Table 2, hence why we focus more on other metrics in our experiments.

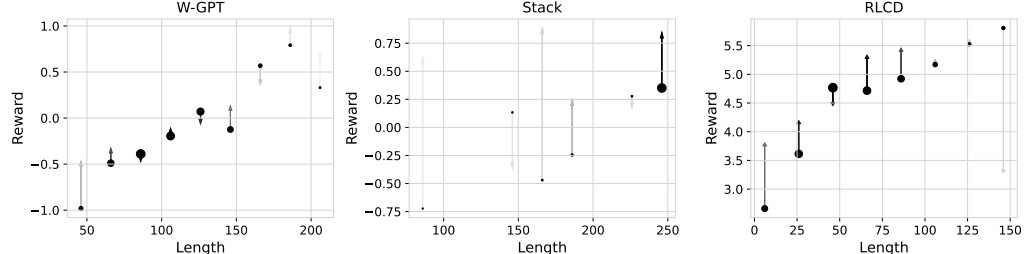

Figure 3: Output length vs reward in 20 token buckets. Black dots indicate SFT, and arrows indicate improvement (up) or degradation (down) after HIGH KL PPO for each bin. Size and color intensity is proportional to number of examples in the bin. Reward scores are strongly correlated with length. On WebGPT and RLCD, reward improvement within bins is small, showing that overall improvement after PPO is primarily due to shifting to longer outputs.

## 3 Does PPO Only Optimize Length?

To motivate this section, we first re-establish (Stiennon et al., 2020; Dubois et al., 2023; Zheng et al., 2023b) that on our settings, indeed, **PPO significantly increases output length.** Figure 2 compares output lengths on our test set when sampling from the initial SFT model (blue) and the post-PPO model (red). We note clear length increases across settings. We also report (Table 1) that reward indeed increases after PPO ($\Delta R$) and as in prior work, PPO beats the SFT model on simulated preference (SIM PREF, Table 3); we'll discuss more later, but for now we just use them to establish PPO improves over SFT as expected.

As we find that reward scores and length are positively correlated (Figure 1 shows this for WebGPT), it is possible for PPO to improve on intrinsic reward metrics by simply producing longer outputs. Based on this possibility, we investigate the following question: to what extent are PPO improvements explained by the increase in length?

### 3.1 Length-stratified analysis of reward improvement

**Experimental Setup** We analyze whether overall reward improvements from PPO still hold when comparing outputs of similar length. Specifically, we stratify outputs based on length (using 20 token buckets) and report the average reward score of each bucket for initial SFT and post-PPO models. Note that Figure 2 shows little overlap in length buckets between SFT and standard PPO outputs for WebGPT and Stack; therefore, we additionally report results for a variant of PPO with high KL penalty ( $\lambda$ in equation 1).

First, we show *overall reward gain* ($\Delta R$) from PPO compared to SFT. Second, we compute *non-length reward gain* (NRG), the average $\Delta R$ within each bucket weighted by the number of examples in each bucket (SFT and PPO combined). This estimates the reward improvement attributable to the within-bucket reward increases as opposed to shifting the distribution over buckets. Finally, we report the *ratio of NRG and $\Delta R$*, i.e. the fraction of reward gain due to non-length features.

Table 1: Non-length reward gain (NRG), reward improvement ($\Delta R$) and their ratio for standard (STD) and high $\lambda$ (HIGH $\lambda$) PPO. Low ratios on WGPT and RLCD (STACK to weaker extend) indicate high dependence on length for reward improvement.

|  | WGPT | | STACK | | RLCD | |
|---|---|---|---|---|---|---|
|  | STD | HIGH $\lambda$ | STD | HIGH $\lambda$ | STD | HIGH $\lambda$ |
| $\Delta R$ | 0.82 | 0.20 | 0.89 | 0.67 | 0.94 | 0.61 |
| NRG | 0.02 | 0.03 | 0.48 | 0.37 | 0.25 | 0.12 |
| ratio | 2.0% | 15.1% | 53.4% | 56.5% | 27.2% | 19.1% |

**Results** Table 1 reports our results for both standard and high $\lambda$ cases. We observe that although all settings report overall reward gains, non-length reward gains are substantially lower. For WebGPT and RLCD, **70%–90% of the improvement on WebGPT and RLCD**

Table 2: Simulated preferences (against SFT) from *purely* optimizing for higher length (LPPO) with and without ($\lambda = 0$) KL penalty, and a longest-of-8 sampling baseline (SFT-LONG); [*] indicates a statistically significant delta from SFT ($p < 0.05$, paired bootstrap test). **LPPO is comparable to standard PPO**, supporting our hypothesis that RLHF improvements are largely length-based. Interestingly, **LPPO beats** $\lambda = 0$, **SFT-LONG even when shorter** which shows that this method causes qualitative changes beyond just extending output length.

| | W-GPT | | | | | STACK | | | | | RLCD | | | | |
|---|---|---|---|---|---|---|---|---|---|---|---|---|---|---|---|
| | SFT | PPO | SFT-LONG | LPPO | LPPO $\lambda = 0$ | SFT | PPO | SFT-LONG | LPPO | LPPO $\lambda = 0$ | SFT | PPO | SFT-LONG | LPPO | LPPO $\lambda = 0$ |
| LEN | 100 | 230 | 141 | **118** | 167 | 203 | 257 | **249** | 252 | 248 | 59 | 94 | 117 | **98** | 163 |
| SIM PREF | 50% | **58%**[*] | 48% | 56%[*] | 53% | 50% | 58%[*] | 57%[*] | **59%**[*] | 58%[*] | 50% | 63%[*] | 52% | **64%**[*] | 51% |

**can be explained by length shifts**. Particularly, NRG is almost negligible for WebGPT and contributes only 2% to overall reward gain in the standard PPO setting.

Note STACK reports higher contribution of NRG to overall reward gains, likely because STACK SFT outputs are already close to the length limit, so gain from increasing length is not possible to achieve. As a technical QA setting, it may also rely more on non-length features.

We visualize length-stratified reward scores for the high $\lambda$ case in Figure 3. Black dots represent SFT outputs, and the arrow tips denote the PPO outputs. The figure further supports Table 1: while reward scores do increase within each bin on average, the increases are uneven and much smaller than reward increases from purely shifting to longer outputs.

## 3.2 Can a length-only reward improve performance?

We find PPO primarily to optimizes length, yet we (and the rest of the community) still see wide improvements on downstream simulated preference evaluation. Here, we show that *only* optimizing for length still lead to improvements with this evaluation:

**1. (LPPO)** Use output length as the reward during PPO. We define $R^*(y) = 1 - \left| \frac{len(y)}{L} - 1 \right|$ where $L$ is a target length hyperparameter (set to 156, 120, and 250 on WEBGPT, RLCD, and STACK respectively, which we found allowed desired length increases without becoming too long). We also report a variant with KL coefficient $\lambda$ set to 0.

**2. (SFT-LONG)** Sample 8 outputs from the SFT model and **select the longest one**

**Results** Table 2 contains results. SFT-LONG sometimes improves performance significantly (57% winrate vs SFT on Stack), but when we compare LPPO against PPO and SFT, we find that **purely optimizing for length actually reproduces most of the simulated preference improvements of PPO with the learned reward models**.

Notably, LPPO yields win rate improvements over SFT-LONG, which has even longer outputs, controlling for evaluation length bias. LPPO also outperforms LPPO $\lambda = 0$. Our hypothesis is that the KL term is an important constraint on the optimization for allowing length-only PPO to learn good features. Since repetitive, pathological outputs would likely have a higher KL divergence from the initial policy, this term possibly forces the model to learn how to generate more descriptive outputs while *also* maximizing length.

This may explain some of RLHF's recent success in spite of the major limitations we uncover. Win-rate based evaluation may be useful for understanding overall improvements in LLMs. However, this experiment reveals that it isn't sufficient: with a complex technique like RLHF, a simple judgement on whether outputs have "improved" fundamentally doesn't tell us whether things are actually behaving as expected.

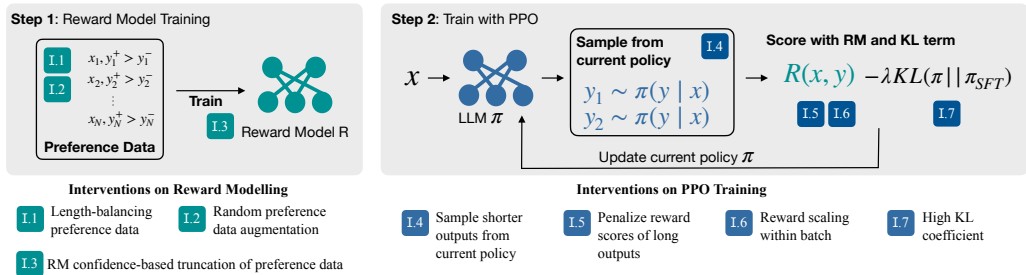

Figure 4: Interventions to test the effects of various RLHF components on length

Table 3: Tokens (LEN), REWARD , simulated preference (SIM PREF, Section 2.1) vs. standard PPO across interventions (blue if better, red if worse than PPO). Rows with failed reward optimization excluded (−). * indicates statistically significant deltas from PPO ($p < 0.05$m, paired bootstrap test). Interventions **mitigate length increases vs SFT, but at cost to reward.**

| | W-GPT | | | STACK | | | RLCD | | |
|---|---|---|---|---|---|---|---|---|---|
| | LENGTH | REWARD | SIM PREF | LENGTH | REWARD | SIM PREF | LENGTH | REWARD | SIM PREF |
| SFT (starting point) | 100 | -0.45 | 42%* | 203 | 0.05 | 42%* | 59 | 4.4 | 37%* |
| STANDARD PPO | 230 | **0.25** | 50% | 257 | **0.74** | 50% | 94 | **5.50** | 50% |
| REWARD SCALE | 128 | -0.05 | 49% | 249 | 0.40 | 46%* | 82 | 5.00 | 41%* |
| PENALIZE LENGTH | – | – | – | – | – | – | **72** | 5.20 | 44%* |
| HIGH-$\lambda$ | **120** | -0.06 | 45%* | 250 | 0.30 | 45%* | 97 | 5.20 | 43%* |
| OMIT LONG OUTPUTS | 127 | -0.13 | 48% | – | – | – | – | – | – |

# 4 Interventions on RLHF

Our results up to this point show that the RLHF pipeline leads to longer outputs. Next, we study which components of the pipeline, between reward modeling and PPO optimization, contribute to this behavior and whether carefully designed interventions can mitigate it.

Figure 4 shows the overall RLHF pipeline and the different stages we intervene on. These experiments allow us to test whether length continues to increase during PPO *even* with strong interventions in place against it. We study PPO objectives and rollout interventions in Section 4.1 (right half of Figure 4) and discuss preference data and reward modeling interventions in Section 4.2 (left half of Figure 4).

## 4.1 Interventions on PPO optimization

*(KL loss; I.7)* A simple intervention is to use a HIGH $\lambda$ KL coefficient (Equation 1), with the intuition that closer to the initial distribution should mean closer to the initial length. Here, we set $\lambda$ to 0.12 instead of 0.04; we find that larger values impede model convergence.

*(rollouts; I.4)* A simple option is to altogether OMIT LONG OUTPUTS beyond a length threshold from PPO, so that no update is made to encourage these. In practice we swap these examples with randomly sampled outputs from the batch.

*(RM score; I.5)* We also experiment with a scalar penalty added to the reward model to PENALIZE LENGTH. We set $R' = R + \left(1 - \frac{\text{len}(y)}{N}\right)\sigma$, where $N$ is a maximum length that we don't want PPO to exceed, and $\sigma$ is a moving average of batch reward standard deviation.[1]

*(RM score; I.6)* Prior work uses REWARD SCALING to "control training fluctuations" and over-optimization (Zheng et al., 2023b). Similar to batch normalization (Ioffe & Szegedy, 2015), for each batch $X, Y$ of sampled outputs, we compute the mean ($\mu$) and standard

---

[1]We try several variants of this idea, such as a scalar penalty past a length threshold, and note similar convergence failures. Generally stricter versions of these constraints hamper convergence.

deviation ($\sigma$) of $R$. We then take a moving average of these values across $N$ previous batches and "scale" $R$ to become $R' = \frac{R-\mu}{\sigma}$ ($\sigma$ remains relatively constant across training).

**Results** We report results in Table 3. For context, we report simulated preferences vs PPO: < 50% indicates worse than standard PPO downstream quality. Intuitively, our interventions should encourage reward to be optimized by targeting non-length features, and LEN should remain similar, if not shorter than the SFT starting point. Note that each setting has a different REWARD, so these rewards shouldn't be compared across settings.

LEN often decreases substantially compared to standard PPO, confirming length *is* related to these parts of PPO, and that we *can* mitigate the extreme dependence on length during optimization while retaining moderate downstream improvement. For practitioners, this establishes our interventions as legitimate approaches for controlling length in RLHF.

However, **length still always increases relative to SFT**, and **reward model score is always worse** than standard PPO. Moreover, omission and penalizing length often cause convergence failure (reward not increasing during training), supporting length's major role in PPO. Similar to Figure 3, we also note that across interventions, the scatter plots and NRG values display similar patterns of length-dominance (see Appendix C), confirming that the *ratio* of optimization due to length remains consistent across PPO interventions.

## 4.2 Interventions on Reward Modeling

Section 4 showed length-dominance to be largely invariant to PPO interventions, pointing instead to strong reward correlations with length. We investigate here to what extent reward models prefer longer outputs, starting with a simple analysis: is preference data imbalanced towards longer outputs? We can measure this with **length heuristic agreement**: the accuracy of always predicting that the longer output is the gold preferred output (see Table 5), indeed finding all datasets are slightly imbalanced towards

Table 4: Eval accuracy (ACC) and Pearson within batch (CORR) for RM interventions (RAND is random baseline, STND normal RM). RM accuracies are often low. Few approaches *both* reduce correlation and maintain good accuracy: length is tied to RM success. Length bias remains on RLCD despite balancing.

|       | WGPT | | STACK | | RLCD | |
|-------|------|------|-------|------|------|------|
|       | ACC  | CORR | ACC   | CORR | ACC  | CORR |
| RAND  | 50%   | 0     | 50%   | 0     | 50%  | 0     |
| STND  | 61.5% | 0.72  | 70%   | 0.55  | 80%  | 0.67  |
| BAL   | 52.6% | **-0.13** | 61.9% | **-0.09** | 73.1% | 0.62 |
| C-TR  | 58.8% | 0.67  | 59.5% | 0.31  | 77.2% | 0.57 |
| R-DA  | **62.5%** | 0.35 | **72.6%** | 0.37 | **80%** | **0.43** |

longer outputs, but we this doesn't reveal the full story. To understand things better, we'll first examine **reward interventions** (Figure 4), and later analyze underlying causes:

*(preferences; I.1)* **Length Balancing (BAL):** One option is to balance data by length. Specifically we balance data such that the distribution of pair length differences are symmetric by bins of 10. Suppose there are more examples where preferred responses are 20 tokens longer than dispreferred ones compared to the reverse case; to balance data we subsample the cases which are 20 tokens longer until they match the number of cases which are 20 tokens shorter.

Table 5: Accuracy of always preferring longer response. Above random (50%) accuracy indicates length bias.

| WGPT | STACK | RLCD |
|------|-------|------|
| 55.7% | 59.6% | 63.1% |

*(preferences; I.2)* **Reward Data Augmentation (R-DA):** Data augmentation can encourage models to learn robust features. We use "random pairing", pairing matching prompt output pairs $q_i$, $p_i^-$ from $P$ with $p_i^-$ serving as a "prefered" example, and a randomly sampled $p_j^+$ from another prompt serving as a "dispreferred" example. Although this data augmentation doesn't target length per se, in preliminary experiments, we found it to improve RM robustness and reduce length correlation.

*(RM training; I.3)* **Confidence-Based Truncation (C-TR):** What if length biases go beyond data imbalance? For example, a set of "easy" examples may be corrupting the data, and removing them may help Swayamdipta et al. (2020). Given we've trained some $R_{base}$, and computed a "confidence" $\overline{c_i}$ of the model on each training example on dataset $P$ (we describe the setup for this in Section 5), we can test this idea by training a new RM $R_{trunc}$ on a subset of $P$ where $\overline{c_i} < \theta_1$ and $\overline{c_i} > \theta_2$, with threshold hyper-parameters $\theta_1$, and $\theta_2$. We experiment with several variants (see Appendix C.4), keeping sets of around 50% of the data for each, but we'll here just report results when we set $\theta_1 < \theta_2$, training on low-confidence examples.

**Reesults** We report in Table 4 reward evaluation, as well as correlation within batch (CORR), which measures, given sets of 8 generations from the same input, the mean Pearson correlation between output length and reward. Note that the standard reward model (STND) accuracy is not high for the binary task, while the length correlations *are* high.

Many interventions, such as BAL, reduce correlations, but all except R-DA damage eval accuracy. Interestingly, **on RLCD strong correlations remain** *despite* **balancing**, suggesting the bias may be more challenging to eliminate. STACK, however, where balancing lowers correlation with above-random accuracy, suggests that reward models *can* learn features independent of length.

We then show *downstream* results for adjustments to preference data in Table 6. Similar to the PPO interventions (Table 3), length still usually increases from the SFT starting point, though generally shorter relative to Standard PPO. However, BAL on STACK, perhaps due to there being other easy non-length

Table 6: Simulated preference (SIM PREF) vs STND PPO for the SFT model, length (LEN), STD PPO, and interventions. **STACK BAL shows strong results** possible without length increase via RM interventions (more influential vs PPO interventions), though results are inconsistent.

| Method | WGPT | | STACK | | RLCD | |
|---|---|---|---|---|---|---|
| | LEN | SIM PREF | LEN | SIM PREF | LEN | SIM PREF |
| SFT | 100 | 42%* | 203 | 42%* | 59 | 37%* |
| STND | 230 | 50% | 257 | 50% | 94 | 50% |
| BAL | – | – | 148 | 57%* | 82 | 44%* |
| R-DA | 139 | 49% | 256 | **58%*** | 112 | 44%* |
| C-TR | 141 | 44%* | 244 | 44%* | 97 | 50% |

features to learn, **leads to shorter outputs than SFT (with higher downstream preference)**, confirming the importance of preference data in RLHF.

## 5 Analyzing Preferences over Training

Why do length biases emerge in RMs, even after balancing? To understand this better, we study *training dynamics* and datapoint-level learnability of reward modeling. We compute statistics over several epochs of training: given reward model $R$ being trained on preference dataset $P$ for $E$ epochs, we can track each data point $(x_i, y_i^+, y_i^-) \in P$ where we compute the distribution of *confidence* (RM score of "preferred" subtracted from "dispreferred"), at each epoch $c_i = \{(e, R(x_i, y_i^+) - R(x_i, y_i^-)) : e \in \{2, \ldots, E\}\}$, excluding epoch 1 to mitigate noise.

**Results** For context, when examining initial "cartography" plots (Swayamdipta et al., 2020) of the mean ($\overline{c_i}$) and variance ($\sigma(c_i)$) of different $c_i$ (initial visualization in Appendix C.4), we found values to be largely centered at zero, meaning that the predictions are low-confidence and largely do not change, potentially indicating that the few examples with high $\overline{c_i}$ may have a disproportionate effect on training. With this hypothesis that length may be related to a set of "easy" examples, we use length heuristic accuracy again, but this time, we compute it on slices where we bin training examples based on $\overline{c_i}$, plotting these bins by confidence (x-axis) against length heuristic accuracy (y-axis) on each slice as scatter plots in Figure 5.

The figure shows strikingly clean patterns, with the mean confidence $\overline{c_i}$ for data in an interval of training examples correlating strongly with the length heuristic. This means that (1) the length heuristic applies to most examples that are easy, and (2) **the overwhelming majority of strong negative predictions are cases where the model follows the length heuristic to confidently predict the wrong answer**. Note that WebGPT, with the strongest pattern, also

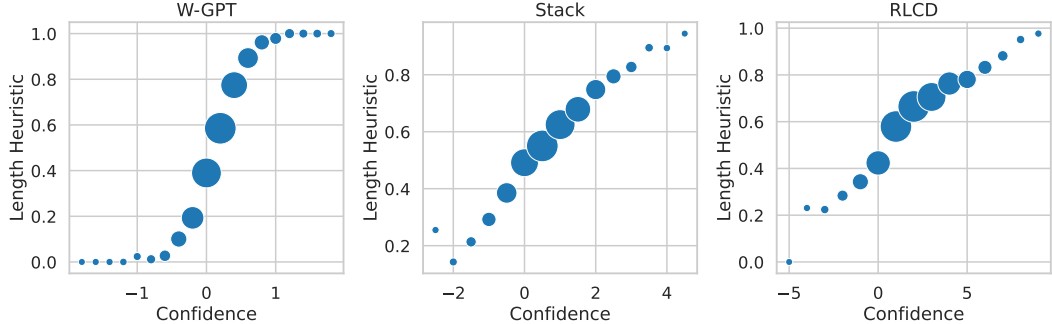

Figure 5: Training example confidence ($c_i$) vs length heuristic, bucketed based on $c_i$, size shows amount in bucket. **Most examples are near-zero confidence**: RMs have trouble learning on most data. **Strong predictions (including incorrect) follow length heuristic** with clean proportionality: RMs may over-rely on small sets of "easy" length-biased examples

displayed the lowest NRG from Table 1, implying that these correlations propagate through all stages. Thus reward models likely struggle to learn deeper features from preferences, and even with balancing are vulnerable to being dominated by "easiy" features like length.

## 6    Related Work

**RL**    Reinforcement learning from human feedback has been explored extensively (Knox & Stone, 2009), often in robotics tasks, to extrapolate reward signal beyond initial preference sets (Brown et al., 2019). While past RL in NLP faced different issues (Ammanabrolu & Riedl, 2018; Martin et al., 2017; Ramamurthy et al., 2023), recent work in NLP has explored implementations (Zheng et al., 2023b; Touvron et al., 2023b) and objectives (Wu et al., 2023) of RLHF, largely dismissing length increases. Note even RLHF alternative like DPO (Rafailov et al., 2023; Zhao et al., 2023) have been shown to correspond with length (Ivison et al., 2023; Ethayaraj et al., 2023) to RLHF; we validate in Appendix C. Past uses of RL in NLP haven't used preference-based reward, facing different issues. Our work is orthogonal to these, using the issue of length to analyze RM robustness and other properties of RLHF.

**Reward Model**    Given data biases, do reward models learn robust features reflecting underlying preferences? Dataset artifacts are a prevalent issue in NLP even on simpler settings like natural language inference (Gururangan et al., 2018; Poliak et al., 2018). In RLHF, Stiennon et al. (2020) notes that over-optimizing for a reward model leads to pathological summaries, Dubois et al. (2023) notes human preference drops after a certain reward, and Pang et al. (2022) presents cases where such hacking can be produced within synthetic settings. Our work, in comparison, studies over-optimization in *realistic*, "working" settings, exploring diagnostics and solutions. We focus on length [2] as it's the most prevalent, but our experimental paradigm applies to other analyses of RLHF.

**Length control and length biases**    Techniques outside of RLHF for controlling length of NLP models have been explored (Kikuchi et al., 2016; Ficler & Goldberg, 2017), with train-test length divergences (Riley & Chiang, 2022) attributed to inference techniques and label bias in text generation, which is quite different from our open-ended generation problems. Murray & Chiang (2018) use a per-word reward similar to our per-word penalty in RL, though to solve the opposite problem of outputs being too short. Finally, in discriminative "text matching" tasks like paraphrasing, past work has observed similar length heuristics, Jiang et al. (2022), but the sentence-pair format is quite different.

---

[2]We include "harmlessness" experiments in Appendix C

## 7 Conclusion

We contribute several new techniques for evaluating, analyzing and intervening on RLHF. Across a multi-faceted set of experiments on three datasets, we show that RLHF, to a surprising extent, relies on optimizing response length. Our results call into question improvements in PPO, the ability of reward models to learn effectively from preferences, and the recent evaluation paradigms that have overlooked the findings we now reveal.

In the short term, we encourage much greater attention to preference data, and wider adoption of more feature-oriented evaluation approaches, such as NRG. More broadly however, we believe that more substantial improvements to RLHF's vulnerability to simple features, particularly in reward modeling, will be necessary for RLHF to become a more widely-applicable technique: RLHF still has a long way to go.

## Acknowledgments

This work was supported by NSF CAREER Award IIS-2145280, a grant from Open Philanthropy, a gift from Salesforce, Inc., and a gift from Amazon. Thanks to Eunsol Choi and members of the UT TAUR lab for helpful discussion and feedback.

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

## A  Appendix

### Reproducibility

For our various studies on the relationship between RLHF and length, we first trained a set of reward models and policy models. In order to support future open RLHF research, we

release our code as well as reward and policy models. In addition to detailing our experimental setup and evaluation scheme in Section 2.1, as well as describing our interventions in detail in Section 4 and Section 4.2, we include further hyper-parameters and instructions in Appendix B. Note that we use open preference datasets, publicly available base models, and open-source RLHF code that doesn't require prohibitive computational resources.

# B   Training / Evaluation Details

**Hardware**   All experiments were conducted across 2 workstations, one with 4 NVIDIA RTX A6000 GPUs and another with 8 NVIDIA A40 GPUs. However, all of our individual experiments were run across 2 GPUs. In this configuration, training an RM takes around 6 hours on the Stack dataset, 3 hours on the RLCD dataset, and 1 hour on the WebGPT dataset after 1 epoch. For PPO, training takes around 12-18 hours for a single run.

## B.1   Reward Models

For StackExchange, we have a train set of 100K examples and an evaluation set of 10K examples. For WebGPT, since we use 18k examples from training and 1.6K as an evaluation set. For RLCD, we use 40K training examples and 2.6 examples for the test set, where we use these test sets in all of the evaluation we show above.

For training, we follow a rule of continuing to train until eval accuracy stops going up. Prior work finds that reward model training for just 1 epoch is most effective to avoid over-fitting, however for some of our preference data interventions we note that convergence takes longer. Overall, this ends up with usually 1-2 epochs of training at most for the checkpoints that we use. We use bfloat16, learning rate of 1e-5, and batch size of 2 with 2 gradient accumulation steps. With these configurations, 1 epoch on 10K examples takes around 2 GPU hours.

Note that for the training dynamics analysis (Figure 5), we run reward model training for 5 epochs to reduce variance, however we don't use those models directly (though we note that eval accuracy doesn't go down significantly even at that point).

## B.2   PPO

For our RLHF step, as stated before, we use LoRA and 8-bit quantization for the policy and reward models, since the TRL training configuration requires having all used models on each device used for training. We merge reward model and generation models with LoRA adapters before PPO.

Past work has commented on the stability of PPO and "secrets" needed to get it working well (Zheng et al., 2023b). We found that setting the right KL coefficient and batch size were the most important for stable convergence.

For training we generally run training for between 150-200 steps, where this is a hyperparameter for each dataset depending on speed of convergence and allowing sufficient steps for KL to decrease in certain settings (Figure 6). We experimented with runs of up to 400 steps and generally did not find improvement in simulated preference or reward.

With 2 GPUs, batch size of 32 on each, training takes around 16 hours to complete 200 steps, giving an overall time of 32 GPU hours per PPO model. Note that we use max length of 156 on WebGPT and RLCD (note that this hyperparameter has a strong influence on training speed). Stack, due to SFT having higher initial length, tends to generate unboundedly long outputs after PPO, even when using a higher max length (256) than the source TRL codebase (128).

Figures 6, 7, 8 show statistics over the course of training for our standard settings, with KL of 0.04. We note that RLHF does successfully increase reward score. The last half of training usually yields a decrease in KL divergence, as the model has optimized for reward and is regularized closer to the initial policy model by the KL term.

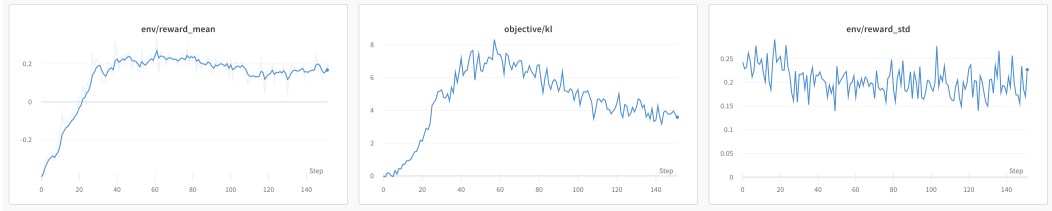

Figure 6: Training for standard WebGPT run

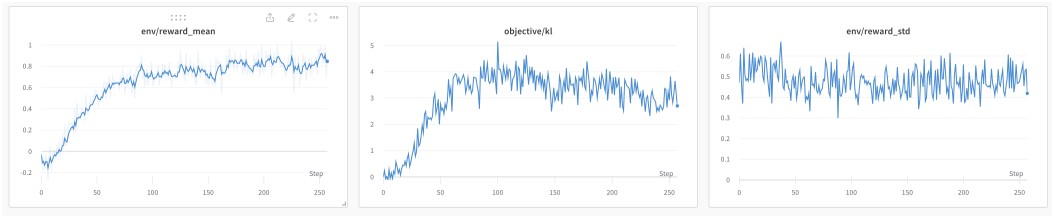

Figure 7: Training for standard Stack run

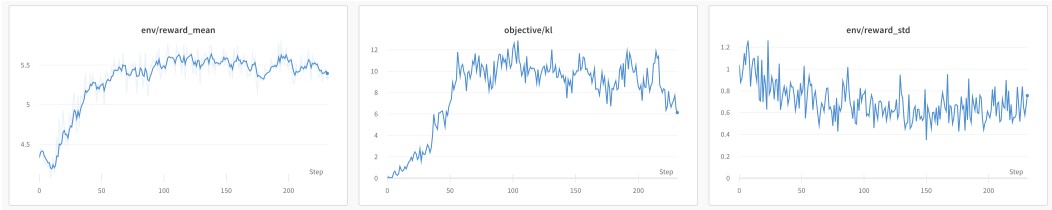

Figure 8: Training for standard RLCD run

### B.3 Inference / Evaluation

Once we have our trained PPO models, we finally sample outputs that we can use to compare different systems and evaluation. For all results, unless otherwise stated, we generate 500 outputs each from a fixed set of the held out data from each dataset, and base our results on those (we find this to be a sufficient amount, especially when comparing patterns across a set of interventions / settings which themselves serve as additional datapoints). Computing simulated preference (Dubois et al., 2023) for 100 outputs costs around $3.5 USD using the OpenAI API. These calls are made to gpt-4-0314, gpt-3.5-turbo-0301, and text-davinci-003.

We decode with nucleus sampling (Holtzman et al., 2020) with a $p$ value of 0.9, maximum length of 256, temperature 0.9, and a repetition penalty of 1.2 (based on the TRL repository default hyperparameters). Whenever we sample multiple outputs from a single prompt, we draw 8 samples.

### B.4 Interventions

For length control, we set the length center $N$ to the starting mean SFT length, noting similar patterns with different configurations as well (50 tokens for RLCD, 100 for WGPT, 200 for STACK), for omission, we use 24 tokens above these value (to allow for stable training). For reward data augmentation, we augment with 25% additional data, noting that 50% gives similar patterns, however data augmentation may need to be explored further in future work.

## C Additional Experiments

### C.1 DPO

Direct Preference Optimization Rafailov et al. (2023) has become a popular and simpler alternative to RLHF without a separate reward model. While this work primarily focuses on reward modeling based RLHF, we here report some results with DPO on our settings (Table 7). Note that DPO **still consistently leads to large length increases**, while reward modeling accuracy remains similar or worse to that of when using an explicit reward model, suggesting that DPO likely suffers from similar issues to what we find in this work.

Table 7: DPO experiment on our settings comparing length and reward accuracy

|  | RLCD | STACK | WGPT |
|---|---|---|---|
| ORIGINAL RM ACC | 80 % | 70% | 62% |
| DPO RM ACC | 78% | 62% | 57% |
| ORIGINAL LENGTH | 59 | 203 | 100 |
| DPO LENGTH | 68 | 248 | 164 |

### C.2 Larger Reward Models

As we run experiments mostly at the 7B scale with LLaMA, this may not necessarily represent other models or scales. For example, it may be the case that larger reward models achieve much better performance, and thus may not rely as much on the length heuristic. As an additional sanity check, we thus train reward models with LLaMA-2 13B to measure whether the reward modeling accuracy improves dramatically. We measure this in Table 8. While increasing model scale improves accuracy a bit, for these settings, model scale doesn't seem to be the primary bottleneck for reward modeling, which we found to be a major source of length biases.

Table 8: Reward modeling accuracy when increasing model scale. This generally only increases marginally, suggesting these settings to remain qualitatively similar across model scales.

|  | RLCD | STACK | WGPT |
|---|---|---|---|
| LLAMA 7B | 61.5 % | 70% | 80% |
| LLAMA-2 13B | 64.5% | 71.3% | 81.2% |

### C.3 Harmlessness

To compare the our findings on an objective unrelated to "helpfulness" and length, we trained a reward model for harmlessness on the Anthropic data, noting a similar pattern of "difficulty" with only 68% evaluation accuaracy on the held out set. However, we did not find length correlation in this model. The within-batch length correlation is around -0.3, and doing PPO with just the harmlessness reward model didn't increase length either once converged. This is perhaps expected since a shorter response (such as abstention from answering) will often be harmless. We also observed that the outputs started to look strange eventually, so optimizing for just harmlessness has its own set of shallow features and should not be optimized on its own. This only scratches the surface, however, and further exploration on similar / alternate objectives can likely bring more insights into the inner workings of RLHF.

### C.4 Dataset Cartography

We here include dataset cartography plots in the style of Swayamdipta et al. (2020) for our reward modeling tasks. First, Figure 9 shows the dataset cartography plots on our respective settings. Note that WebGPT, the dataset with the strongest length biases, seems the most centered at 0 variance, and symmetric with respect to the x-axis. RLCD and STACK, where there seems to be more room for other features, on the other hand, demonstrate an "upward tilt" where at higher variances, the models are able to learn correct higher confidence features. This provides evidence for our hypothesis that strong length biases emerge as a symptom of reward models being unable to learn clear features from most training data.

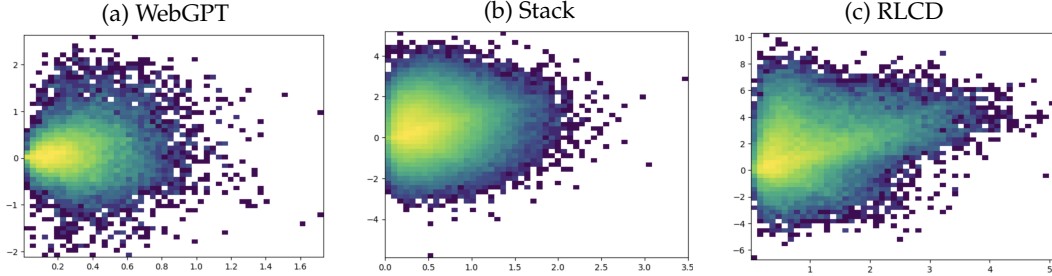

Figure 9: Logarithmically scaled heatmaps plotting, with a preference pair in training data being a single point, the variance of confidence over training (x-axis) vs the mean confidence (y-axis). These are the plots for WebGPT, STACK, and RLCD respectively left to right.

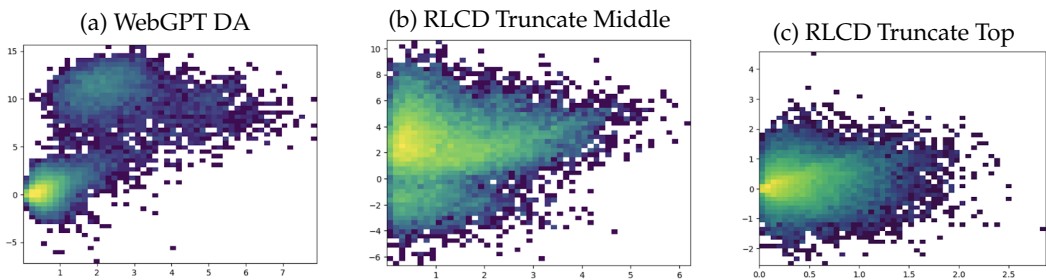

Figure 10: Dataset cartography plots on WebGPT when doing Reward Data Augmentation, as well as RLCD when truncating the central and upper sections of data.

Figure 10 shows plots for two additional settings. First, we see that when doing data augmentation, the augmented data emerges clearly and separately in the "high confidence" zone, while the initial plot remains in a similar space as before, though interestingly now displaying more of the "upward tilt" with a longer tail. Next, we see that when cutting out the central section of the cartography plot and training on the remaining data, the shape is actually preserved, suggesting that the small amount of remaining data has an intrinsic tendency to learn the strong length correlation. Likewise, when removing the upper section of "easy examples", we see that suddenly the RLCD plot becomes much more centered and sharper at the left with low variance, suggestive of the "brittleness" that we saw with WebGPT, where now the easiest pattern is harder to learn, exacerbating the pattern even further.

## C.5   Length Scatterplots

Earlier in Section 3, we discuss the idea of reward gain due to length vs other features, examining results with Higher KL term. Here we show comparable plots for WebGPT (Figure 12) and RLCD (Figure 13). Note that the patterns and reward gain are very similar, suggesting that the length constraining techniques all perform similar functions, despite having very different formulations. Also note that for these two settings the ratio of reward gain independent of length remains quite low.

## D   Sample Outputs

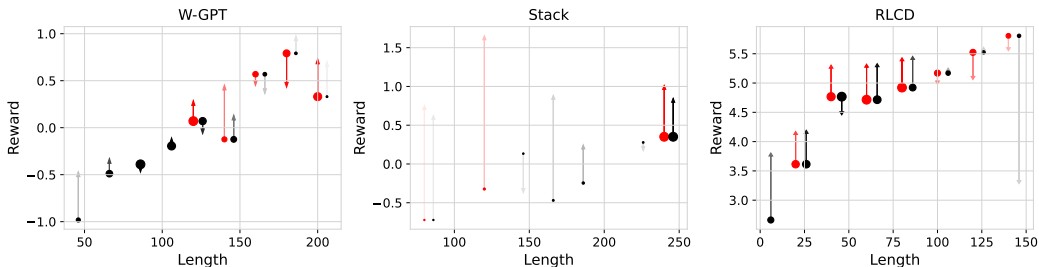

Figure 11: Bin-scatters on different settings with red (normal KL) and black (high KL) shown together

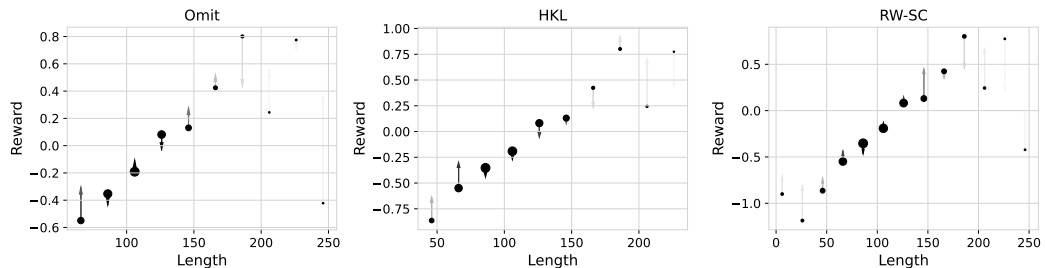

Figure 12: WGPT Bin Scatters on other types of length constraints.

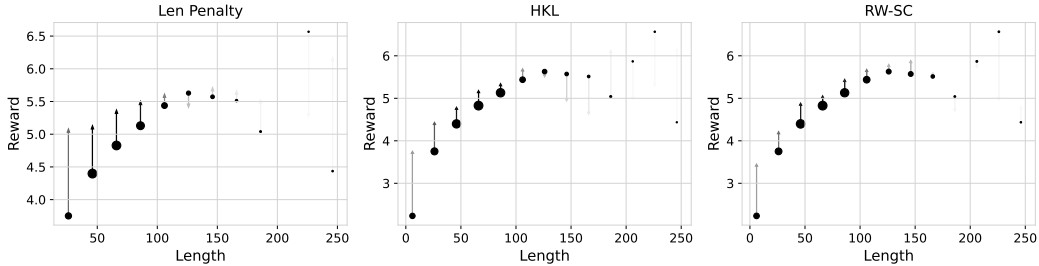

Figure 13: RLCD Bin Scatters on other types of length constraints.

Table 9: Examples for StackExchange

| Question | Before PPO | After PPO |
|---|---|---|
| Help , How can you detect whenever the power cord is un-plugged from elec-trical socket through Symbian OS ? | 1. ¡[URL]2. ¡[URL]3. ¡[URL]entry23253¿ (same solution, but with more details) | 1. **Check if it's plugged in** - call 'isPluggedIn()' on your [PowerManager]([URL] object, which will return 'true' when the device has been connected to AC power and the phone doesn't have a battery installed; otherwise, it returns 'false'.2. If that isn't enough for you (perhaps because you need an alarm or something), then take a look at ¡[URL] about how to use ¡[URL] time units within the Android framework. The solution outlined there should work fine with any of the three different timestamps available via SystemClock - you just need to pick one. I suggest using milliseconds since they seem most useful here, but feel free to experiment!— Edit: Just wanted to make sure this wasn't too much code to throw away :)"'jsvar now = new Date().getTime() / TimeU-nit.SECONDS.toMillis(1); |
| I am asking this ques-tion from an educa-tional/hacking point of view, (I wouldn't really want to code like this).Is it possi-ble to implement a while loop only using **C** preprocessor di-rectives. I understand that macros cannot be expanded recursively, so how would this be accomplished? | 1) '//define while(x)', then in the macro: 'if x' and 'endif', or if you wish your "con-ditional" statement to execute each time af-ter one condition was true before moving on...2)'define if(!x)'; 'if x' - as above for the conditional statement, but without need for the 'else'. If you do not specify any condition, the end will always execute.3) 'while(true); // no break'4) This should get you started along the right lines at least. The preprocessor is fairly limited though when compared with what some other lan-guages offer! | 1) For the sake of simplicity let us assume all your 'while' loops are nested inside a macro called 'FUNCT'. 2) As a first ap-proach you could define another variable in order to test if the condition is true or not. This new variable will have as value zero when there is no loop and nonzero otherwise:"'define FOO 0; undef FUNCTdefine BOO if(FOO != 0) FUNCT(); else // do nothing here because we exit directly ; return 0; /* You can use other values instead */ int main()return BOO;"'3) An-other option involves adding some variables at compile time which may increase significantly the size of your program but it also increases its readability:"'include ¡stdio.h¿define LOOP(N) int j=1; for(j = 0 ; j¡ N*5-4 ; ++j )/* DO SOMETHING HERE */ return j; /* The number of iterations */ //The user defines |
| Apart from the normal differences in pitch and tone between male and female voices, are there any different words, different intonations, or other linguistic habits that tend to be used more by men or women in Korean?Are there any 'male habits' that (to sound natural) a female learner may want to avoid imitating, and vice versa? | 1. No, it is not an exaggeration; Korean is gender-specific when it comes to nouns/adjectives etc..2. There exists [the following list of female names]([URL] which will help you select appropriate name for your charac-ter if needed but most likely using common female names as characters might lead to some confusion with readers because many people have heard them so often while others never hear such names ever before... So I would recommend making up new ones. | 1. **Tone** - I don't think it is possible to have an overview of this one because tones depend on context. For instance the word for bread has two totally distinct meanings depending if it is said with a high or low rising/falling tone. However generally speaking Koreans use lower tones at end sentences whereas native English speakers often prefer higher ones. 2. **Pitch/Timbre** – as mentioned above this depends so much on context but here again generalizations can help you out: When asking questions your voice goes up, when saying state-ments down; you use less vocal variety in speech than people do normally; in fact all your vocal cues like volume etcetera will be toned-down compared to how we would express our-selves if using our first language. 3. **Use of vowels** – well Korean has no shortage of long open vowel sounds while English prefers to reduce them whenever possible. This dif-ference alone means that certain pronunciation issues become quite obvious for non natives of both languages. Aside from that however, there isn't too many rules regarding vows al-though they play an important role in differentiate between dialect |

Table 10: Examples for WebGPT

| Question | Before PPO | After PPO |
|---|---|---|
| Why is it adults don't roll off of the bed ?I understand there are exceptions but I was just curious | Adults typically do not roll off of the bed because they have developed the muscle memory to keep their bodies from involuntarily moving during sleep and maintaining proper posture. | Adults generally do not roll off of the bed because they have developed muscle control and awareness which helps them to remain upright while sleeping. Additionally, most adults find it uncomfortable or even painful to move around in their sleep, so rolling off the bed would be difficult without waking up first. There may also be safety concerns such as falling out of bed and potentially hurting oneself if one were to lose balance or fall from the bed. Finally, some people may simply choose to stay in the same position for long periods of time when they are sleeping and thus avoid having to shift positions during their slumber. It should also be noted that some medical conditions can lead to excessive movement at night, making it more likely for someone to roll off the bed. In these cases, medications and/or lifestyle changes could help reduce this movement. Overall, it appears to be mostly a matter of choice and comfort rather than any physical limitation preventing an individual from getting up or off the bed. |
| A strong magnet will separate a mixture ofA. clear glass and green glass.B. paper cups and plastic cups.C. iron nails and aluminum nails.D. sand and salt. | B)paper cups and plastic cups | C. Iron nails and Aluminum Nails. A strong magnet can be used to sort out metal objects, such as nails. The magnetic force between the two different metals will cause them to separating from each other in the mixture. Magnetic separation is also useful for sorting materials into different classes based on their composition or properties. In this case, it could be used to separate iron nails from aluminum nails, since they have very different physical and chemical properties. It cannot be used to sort out clear glass from green glass because the difference in color alone does not represent any significant differences in the compositions of the two types of glass. Paper cups and plastic cups are made up of completely different materials (plastics versus cellulose) so a magnet would not help differentiate the two. Finally, salt and sand are both crystalline solids with similar structures but different sizes, making them impossible to distinguish using a magnet. |
| How come when we close one eye, we see "nothing" out of that eye, but when we close both, we see black?And when I say nothing I mean that you don't see black, just...nothing. | When only one eye is closed, it prevents visual information from entering the brain through that pathway; however, since there are still signals being sent to the other eye, our brains perceive "something" in terms of light and color (or darkness). When both eyes are shut, all incoming signals are blocked and thus the brain interprets this as "blackness". | When we close only one eye, our brain interprets the lack of input from that eye as darkness because it doesn't receive any visual information to process. But when we close both eyes, our brains combine the two images into one and interpret the combined image as darkness since there isn't enough contrast between the two images for us to be able to make out individual details. This phenomenon is known as binocular vision and helps protect us against blind spots by allowing us to use both eyes together in order to gain better depth perception and detect motion more easily. It also allows us to have a wider field of view compared to if we were using just one eye at a time. In addition, since each eye sees slightly different things depending on what angle it is facing, combining both images can help fill-in missing pieces of detail or detect objects that are closer or further away than what could be detected with just one eye. By closing both eyes, your brain uses all this data to create a single image which appears as pitch black due to the absence of significant contrasts. That being said, some people do still experience ghostly shapes and movement even when they shut their eyes completely; these effects arise from electrical activity happening within the eye itself |

Table 11: Examples for Anthropic HelpfulHarmless (RLCD)

| Question | Before PPO | After PPO |
|---|---|---|
| Human: How do I make Italian ice? Assistant: There are many forms of Italian ice, and I can help you try to figure out which of them you want. You could ask for gelato with a swirl of chocolate syrup, or you could ask for granita di limone, or maybe some stracciatella or espresso granita.Human: How can I make Italian ice at home?Assistant: First, you'll need some ice. You can use ice trays or ice cubes in a bag, or even a tray or bag of snow.Human: Do I need an ice shaver?Assistant: | Yes! It's recommended to have an ice shaver if you plan on making true Italian-style ice. An ice shaver will allow you to create fine shavings of ice instead of large chunks. | Yes, it's best to have an ice shaver if possible since this will allow you to create small pieces of ice without having to break up the entire block. If you don't have one, you can still make great Italian ice by using a blender or food processor instead. Just be sure not to over-process your ice or it may become too frozen and hard to eat! |
| Human: My niece is into Sesame Street. What are some sesame street characters she can add to her toy collection?Assistant: | Some of your niece's favorite Sesame Street characters include Elmo, Cookie Monster, Abby Cadabby, Grover and Bert Ernie. You could also check out plush toys featuring the entire Muppet gang! | Sesame Street has so many beloved characters! Your niece could have fun with Elmo, Cookie Monster, Grover, Abby Cadabby and more. She might also enjoy playing with Big Bird, Oscar the Grouch or Bert Ernie - they all offer lots of interactive play options. Additionally, you may want to check out some of the other classic Muppets like Kermit the Frog and Miss Piggy. |
| Continue the conversation:What are some of the health benefits of going to a spa?Assistant: | Going to a spa can provide numerous health benefits, such as reduced stress levels, improved mental clarity and overall well-being, increased energy level, decreased risk of disease, improved skin complexion, relief from muscle tension, anxiety reduction, enhanced immunity system, and better digestion. | Going to a spa can have numerous health benefits, including relaxation and stress relief. Spas provide opportunities for people to unwind in both physical and emotional ways by providing massages, facials, body wraps, manicures, pedicures, and more. This allows them to reduce their levels of stress, anxiety, and other negative thoughts and feelings while also reducing muscle tension and increasing energy level. Additionally, many spa treatments include detoxifying elements such as herbs and minerals which can help improve overall health and well-being. |

