# OpenReview forum: "A Long Way to Go: Investigating Length Correlations in RLHF"
_colmweb.org/COLM/2024/Conference — COLM_

### Official Review · Reviewer_zdy6 · 2024-05-04

**Rating:** 8
**Confidence:** 4
**Ethics Flag:** 1

**Summary:**

This paper reveals an issue regarding the verbosity of large language models (LLMs). Specifically, the authors demonstrate that the response length is a significant factor behind RLHF. The authors further find that even a purely length-based reward reproduces most downstream RLHF improvements over supervised fine-tuned models. Testing a comprehensive set of length-countering interventions, this paper identifies the dominant source of the biases for the reward models. The main contribution of this paper is to reveal the vulnerability of the RLHF process.

**Questions To Authors:**

N/A

**Reasons To Accept:**

I think this paper does an excellent job of identifying the bias regarding the output length of LLMs.

**Clarity.** This paper is well-written and organized. I think this paper is easy to follow.

**Importance.** Evaluation of LLMs is a very timely and important topic in the field of AI. Verboisity bias has been highlighted in several concurrent works (please see the specific paper titles in the Reason to Reject section). This paper investigates such bias in LLMs in detail. I think the empirical analysis of this paper is quite convincing, and there is little room for objection.

**Empirical experiments.** I think the empirical experiments are well-designed. The results are consistent with other papers, and I think they are reasonable.

**Reasons To Reject:**

I don't come up with a strong reason to reject this paper, but I think there are a few missing citations regarding length or verbosity bias in RLHF/DPO. These papers can be regarded as concurrent work, so it is optional for authors to refer to them.

- Park, Ryan, et al. "Disentangling length from quality in direct preference optimization." arXiv preprint arXiv:2403.19159 (2024).
- Saito, Keita, et al. "Verbosity bias in preference labeling by large language models." arXiv preprint arXiv:2310.10076 (2023).
- Wang, Haoxiang, et al. "Arithmetic Control of LLMs for Diverse User Preferences: Directional Preference Alignment with Multi-Objective Rewards." arXiv preprint arXiv:2402.18571 (2024).

---

> ### Author Rebuttal · Authors · 2024-05-28
>
> Thanks for the thoughtful review! We will add discussion of those citations you mention in any future version.

---

> > ### Comment · Reviewer_zdy6 · 2024-06-04
> > **Acknowledgement**
> >
> > I've read other reviews and authors' rebuttals. I did not change my mind and will keep the original score.

---

### Official Review · Reviewer_WB4w · 2024-05-11

**Rating:** 6
**Confidence:** 4
**Ethics Flag:** 1

**Summary:**

This paper explores the tendency of Reinforcement Learning from Human Feedback (RLHF) to produce longer outputs when aligning LLMs with desired properties, such as helpfulness in dialogue and web question answering tasks. The authors delve into the strategies RL optimization employs to maximize reward and find that improvements are often driven by increasing response length rather than other features.

**Questions To Authors:**

- Would developing more capable reward models benefit this issue? With limited GPU resources, the authors could at least evaluate whether top-ranked models on RewardBench (e.g. Eurus-RM-7B and Starling-RM-34B) still prefer longer responses.

**Reasons To Accept:**

- The length bias issue is worth a deep investigation. The final conclusion of reward modeling is well-supported by experiments.
- The controlled experiments on length-only reward clearly show length shortcuts.

**Reasons To Reject:**

- Would the length bias issue correlate with tasks? The selected datasets are question answering and chatting, which may naturally bring the length bias issue. I suggest selecting some length-unrelated tasks such as math, coding, or length-constrained instructions for a more comprehensive evaluation.
- In my opinion, before intervening in the RLHF process to mitigate the lengthy outputs, it is more important to recognize whether human annotators and LLM judges commonly prefer lengthy outputs, and when it is a bias.

---

> ### Author Rebuttal · Authors · 2024-05-28
>
> Thanks for the thoughtful comments! We’ll try to address concerns here.
>
> > more capable models
>
> We conducted additional experiments with scaling up our RM experiments up to LLaMA-2 13B. We report intrinsic reward modeling accuracy results below (compare with Table 4 in the paper):
>
> | Model | WebGPT | Stack | RLCD |
> |-------|--------|-------|------|
> | LLaMA 7B    | 61.5%  | 70%   | 80%  |
> | LLaMA-2 13B   | 64.5%  | 71.3% | 81.2%|
>
> On these datasets, reward modeling accuracy is only marginally better than it was before, suggesting that increasing model scale isn’t necessarily the main bottleneck on these tasks. While the settings are a bit different, we will definitely consider extending our study to evaluate other models in future work.
>
> > length-unrelated tasks such as math, coding, length-constrained instructions…
>
> This is a great point. It’s worth noting that we actually already have such a setting (the Stack setting) which we chose because it represents a realistic setting where math, coding and more complex reasoning may be used. Indeed, we do note this setting to demonstrate most of the discussed length-correlated behavior throughout the RLHF pipeline, though to a lesser degree.
>
> > Would the length bias issue correlate with tasks?” “In my opinion, before intervening in the RLHF process to mitigate the lengthy outputs, it is more important to recognize whether human annotators and LLM judges commonly prefer lengthy outputs
>
> We agree with this point and discuss it in the paper (introduction, 3.1): length may be a valid feature corresponding to quality for humans. Two of the datasets we run experiments on, Stack and WebGPT, use human preference labels and these labels do show length biases (Table 5).   However, these are fairly small, suggesting that length may not be a dominant feature for human annotation. These small length biases are not proportional to the dominance of length we find experimentally after RLHF on these same settings, which is particularly concerning given how commonplace these techniques / evaluations are. Regardless of whether length is a good or bad feature, we find it concerning if it comes at the cost of other potential good features.

---

> > ### Comment · Reviewer_WB4w · 2024-06-05
> > **Reply**
> >
> > Thanks for the reply. The additional results are great. I will increase my score.

---

> ### Comment · Area_Chair_4ZkL · 2024-06-04
>
> Hi WB4w, can you check the authors' response and update your review if it addressed your concern (or participate in discussion with the authors if it did not)?

---

### Official Review · Reviewer_Bn2p · 2024-05-14

**Rating:** 6
**Confidence:** 4
**Ethics Flag:** 1

**Summary:**

The paper investigates the relationship between output length and performance in Reinforcement Learning from Human Feedback (RLHF) for large language models. The authors demonstrate across three diverse settings (WebGPT, Stack, and RLCD) that RLHF tends to significantly increase output length, and a substantial portion of the reported improvements in reward can be attributed to this length increase rather than other meaningful quality improvements. Through controlled experiments, the authors show that optimizing for length alone (using a length-based reward) can reproduce most of the gains seen in standard RLHF. They explore various interventions to mitigate the length bias, including adjustments to the reward model training, preference data, and PPO optimization, but find that length biases persist across these interventions. Further analysis reveals that reward models exhibit strong correlations with length, potentially due to overfitting to a small set of "easy" length-biased examples during training.

**Questions To Authors:**

- In page 7, Reesults -> Results.

**Reasons To Accept:**

- The paper provides a comprehensive analysis of the relationship between output length and RLHF performance, which has been noted but not thoroughly studied in prior works. The experimental setup covers diverse settings and tasks, lending robustness to the findings.
- The analysis is multi-faceted, involving controlled experiments, interventions, and in-depth investigation of training dynamics, and the paper proposes practical interventions and evaluation techniques (e.g., NRG) to mitigate and assess length biases in RLHF.
- I think the findings are meaningful and call for more careful consideration of preference data quality, reward model robustness, and evaluation metrics in RLHF research.

**Reasons To Reject:**

- The chosen evaluation datastes are limited and more results on general benchmarks like MT-bench or AlpacaEval 2.0 can make the results more convincing. Also, it would be interesting to see the performance change on evaluations that does not rely on the lengths of the outputs (like MMLU).
- Due to the computational resources, the authors did not conduct experiments on larger models, and the the PPO training is based on LoRA. Even without using a larger model, the authors can try the experimental results on more powerful small models (such as mistral, llama-2 and llama-3).

---

> ### Author Rebuttal · Authors · 2024-05-28
>
> Thanks for the thoughtful comments! We address the main concerns here:
>
> > larger models
>
> We conducted additional experiments with scaling up our RM experiments up to LLaMA-2 13B. We report intrinsic reward modeling accuracy results below (compare with Table 4 in the paper):
>
> | Model | WebGPT | Stack | RLCD |
> |-------|--------|-------|------|
> | LLaMA 7B    | 61.5%  | 70%   | 80%  |
> | LLaMA-2 13B   | 64.5%  | 71.3% | 81.2%|
>
> On these datasets, reward modeling accuracy is only marginally better than it was before, suggesting that increasing model scale isn’t necessarily the main bottleneck on these tasks.
>
> We will add these experiments in any future version.
>
> > The chosen evaluation datasets are limited and more results on general benchmarks like MT-bench or AlpacaEval 2.0 can make the results more convincing
>
> This is a fair point. We aim to cover a fairly diverse set of realistic tasks (multi-turn dialogue QA, single-turn long-form QA, and technical question answering) to demonstrate our findings on publicly available preference datasets at the time the work was conducted. We do note that the pairwise preference metric we report (setting specific) is based on GPT-4 evaluation which is used on MT-Bench and alpaca-eval and thus should give a reasonable approximation to what one may find in those settings, the main difference being the prompt set.
>
> As for MMLU, we think this is a nice suggestion, but the fine-tuning recipes (datasets used, SFT, etc.) necessary to induce good chain-of-thought-based question answering are a bit different than the recipes for the kind of LLM alignment we study here.

---

> ### Comment · Area_Chair_4ZkL · 2024-06-04
>
> Hi Bn2p, can you check the authors' response and update your review if it addressed your concern (or participate in discussion with the authors if it did not)?

---

### Official Review · Reviewer_kVck · 2024-05-23

**Rating:** 9
**Confidence:** 3
**Ethics Flag:** 1

**Summary:**

This paper presents a well-scoped investigation of Reinforcement Learning with Human Feedback (RLHF), examining an observed tendency to prefer longer outputs and how it may be mitigated. The paper is concise and convincing, highlighting an important and overlooked feature of RLHF, by questioning what our preferences actually encourage in the generated outputs. This is an important step toward better evaluation of and methods for alignment techniques such as RLHF.

**Reasons To Accept:**

To study the effect of length in RLHF, the authors investigate a satisfactory span of data: a long form and a short form question answering dataset (WebGPT, Stack), and a multi-turn dialogue dataset (RLCD). They provide sufficient, detailed information about the models and hyperparameters used.

Their argument is well-structured, each step culminating with empirical results. First, the paper presents an empirical justification that there exists a difference in preferred length with Figure 3. Then, an investigation of how much the length explains the reward with Table 1. Finally, a study of what interventions can mitigate rewarding long responses supported by figures and tables throughout section 4.

I believe this is an important paper, because it questions and systematically observes a flaw in a method that otherwise appears ubiquitously applied.

The paper investigates whether and to what degree a popular method, Reinforcement Learning using Human Feedback (RLHF), is susceptible to producing text generations that are longer but not truly better. It is important to question contemporary, widely used techniques, and this paper does so in a principled way. This will be useful as a step towards further critiquing RLHF, and refining alignment methods.

The interventions proposed by the paper are reasonable and comprehensive. They bring clarity to the different parts of the RLHF components and how each can be influenced, and how effective this is.

The paper is clear and concise, explaining their decisions succinctly and motivating them well. Their background section had just enough detail to be relevant without extraneous information. Their claim and findings are clear in the abstract, and I found it easy to keep them in mind while reading the rest of the paper. I did not notice inconsistent reasoning.

**Reasons To Reject:**

I did not understand the last sentence of 2.1, "To control for length bias, we focus on results where shorter outputs show greater win-rates". Specifically, how this was done throughout the paper (is it distinct from showing that shorter outputs are less frequently associated with greater win-rates?) and why this is a representative way to support the argument.

---

> ### Author Rebuttal · Authors · 2024-05-28
>
> Thanks for the thoughtful comments!
>
> > the last sentence of 2.1…
>
> Thanks for pointing this out! We mainly mentioned this to qualify the win-rate preference results, since these may themselves have an undesired length-bias (e.g. STACK on SFT-LONG in Table 2), which has been supported in other recent work as well [1]. We will clarify this more in future versions.
>
> [1] Yann Dubois, Balazs Galambosi, Percy Liang, and Tatsunori B. Hashimoto. LengthCorrected AlpacaEval: A Simple Debiasing of Automatic Evaluators. https://github. com/tatsu-lab/alpaca_eval, 2024.

---

### Official Review · Reviewer_rdjK · 2024-05-24

**Rating:** 8
**Confidence:** 4
**Ethics Flag:** 1

**Summary:**

In this paper, the authors present a thorough investigation into how RLHF tends to optimize for longer output lengths, often at the expense of true quality improvements. They study this phenomenon across three diverse task settings - WebGPT, Stack, and RLCD. On WebGPT and RLCD, the bulk of the reward improvement from RLHF can be attributed simply to increases in output length, with little gain coming from optimizing other features. Attempts to mitigate the length bias through various interventions in the RLHF pipeline, such as adjusting the reward function, policy rollouts, KL loss, etc. do reduce length increases to some degree, but fail to eliminate the bias and often hurt overall performance. Overall, this is a well-executed and insightful study that reveals some concerning flaws in current RLHF approaches. The authors convincingly show that a significant portion of recently reported "progress" may be illusory and simply due to producing longer outputs. The finding that even preference datasets balanced by length still produce biased reward models is particularly notable. I believe this paper is great with its originality and rigorous experiment setting.

**Questions To Authors:**

- For experiment setup in Section 3.1, I wonder is sft output or ppo output you would stratify based on the length, or combined?
- For Table 2, Could you please win/tie/lose rate in addition to win rate only? I believe this could strengthen the performance comparision between models.
- For Table 3, why std ppo has failed reward optimization? As it has length and reward, I think it should be accessed by sim perf also?

**Reasons To Accept:**

- The experiments are quite neat. I truly appreiciate the rigorous experiment pipeline to understand the underlying mechanism for rlhf. The overall 'causal' experiments are quite convincing, make it a strong candidate for the venue.
- Insightful Analysis. Beyond just showing the existence of length biases, the paper provides a detailed look into their causes and the training dynamics involved. The finding that biases stem from the preference data itself and remain even after balancing attempts is an important insight with implications for data collection practices.

**Reasons To Reject:**

I enjoy reading this paper but the only confusing section for me is Sec 3.1. I can not get why non-length reward gain (NRG) is calculated as average ∆R within each bucket weighted by the number of examples in each bucket. What does this stand for? For me the most intuitive way to investigate the correlation between reward gained by ppo and its length is just to calculate $∆R/∆L$, where $∆L$ is the length difference between sft model and ppo model.

---

> ### Author Rebuttal · Authors · 2024-05-28
>
> Thanks for the thoughtful comments! We’ll try to address the questions below:
>
> >I enjoy reading this paper but the only confusing section for me is Sec 3.1… most intuitive way is to just calculate ∆𝑅/∆𝐿
>
> This is a great question! We’ll try to clarify here.
>
> Intuitively, we wanted to compute reward increase $\Delta R$ when $\Delta L$ = 0. Because these cases are quite rare, our approximation of this is to compute the $\Delta R$ within each length-stratified bucket, as $\Delta L$ within buckets is quite small. We take an average of this across buckets, weighted by the number of instances (SFT + PPO combined) in each bucket.
>
> We like the suggestion of reporting $\Delta R$/$\Delta L$ or correlation between $\Delta R$ and $\Delta L$! One slight limitation of $\Delta R$/$\Delta L$ is that it changes the units and is therefore not comparable to $\Delta R$ in the table. Furthermore, if we computed $\Delta R$/$\Delta L$ (assuming you meant $\sum_i (\Delta R) /  \sum_i (\Delta L)$), it could be the case that length increases and reward increases, but these are unconnected. Controlling for length with our approach helps disentangle this. Note that we already report corr(𝑅, 𝐿) within a batch in Table 4.
>
> > For experiment setup in Section 3.1, I wonder is sft output or ppo output you would stratify based on the length, or combined?
>
> We stratify based on length here, where we compare all the outputs from SFT that are in some length range (e.g. 20-40 tokens) with all the PPO outputs in the same length range (20-40 tokens). Note that the overall prompt set is the same, but each bin may not have comparable prompts between SFT and PPO.
>
> > For Table 2, Could you please win/tie/lose rate in addition to win rate only? I believe this could strengthen the performance comparison between models.
>
> Our prompts (same as AlpacaFarm) ask GPT-4 to return one model name as output, corresponding to the better output. Because of this we rarely observe ties, thus the lose rate is just the reverse of win-rate.
>
> > For Table 3, why std ppo has failed reward optimization? As it has length and reward, I think it should be accessed by sim perf also?
>
> Thanks for pointing this out! Standard PPO doesn’t have failed reward optimization; we used a - in the sim pref column for it because it is the reference point. We can replace those numbers with 50% for better clarity.

---

### Comment · Area_Chair_4ZkL · 2024-06-02
**Discussion period is now open**

Hi reviewers, please take a look at the author's rebuttals and the other reviews for this paper!

If the rebuttals addressed your concerns, please let the authors know about this and update your review. If not, please continue to engage with the authors and the other reviewers in the discussion forum.

Three of our five reviewers are very positive about this paper, and Bn2p and WB4w are less positive.

Bn2p:
- One of your concerns was that experiments were not performed on larger models; however, the authors conducted these experiments during the rebuttal period. Does this address that concern?
- You also were concerned about the choices of datasets. Do you find the authors' response addresses this concern?

WB4w:
- Your question about different models is also addressed by the authors.
- Your concerns about length bias correlation with tasks, as well as concerns about human preferences, also received a response from the authors. Do you find this addresses your concerns?

For the other reviewers, would you like to argue for this paper's acceptance?

---

### Decision · Program_Chairs · 2024-07-10

**Decision:**

Accept

**Comment:**

This paper investigates the influence of RLHF on features of LM-generated text, particularly focusing on how it influences length of generated responses (in particular, causing an increase in length). The experiments suggest that reward improvements are primarily due to increases in length, e.g., by using response length as a reward and finding most improvements from RLHF are retained after RL-based fine-tuning, and find that this is mostly due to trained reward models learning a strong length bias (i.e., assigning higher reward to longer responses).

The experiments are well-designed, and the insights this paper provides are really important given that RL-based finetuning (especially RLHF) is currently the dominant paradigm in training LMs.

I suggest the authors incorporate feedback and suggestions from the reviewers.